# Optimizing HIV Patient Engagement with Reinforcement Learning in Resource-Limited Settings

África Periáñez
africa@causalfoundry.ai
Causal Foundry
Barcelona, Spain

Kathrin Schmitz
kathrin.schmitz@m2m.org
mothers2mothers
Cape Town, South Africa

Lazola Makhupula
Lazola.Makhupula@m2m.org
mothers2mothers
Cape Town, South Africa

Moiz Hassan
moiz@causalfoundry.ai
Causal Foundry
Barcelona, Spain

Moeti Moleko
moeti.moleko@m2m.org
mothers2mothers
Cape Town, South Africa

Ana Fernández del Río
ana@causalfoundry.ai
Causal Foundry
Barcelona, Spain

Ivan Nazarov
ivan@causalfoundry.ai
Causal Foundry
Barcelona, Spain

Aditya Rastogi
aditya@causalfoundry.ai
Causal Foundry
Barcelona, Spain

Dexian Tang
dexian@causalfoundry.ai
Causal Foundry
Barcelona, Spain

## ABSTRACT

By providing evidence-based clinical decision support, digital tools and electronic health records can revolutionize patient management, especially in resource-poor settings where fewer health workers are available and often need more training. When these tools are integrated with AI, they can offer personalized support and adaptive interventions, effectively connecting community health workers (CHWs) and healthcare facilities. The CHARM (Community Health Access & Resource Management) app is an AI-native mobile app for CHWs. Developed through a joint partnership of Causal Foundry (CF) and mothers2mothers (m2m), CHARM empowers CHWs, mainly local women, by streamlining case management, enhancing learning, and improving communication. This paper details CHARM's development, integration, and upcoming reinforcement learning-based adaptive interventions, all aimed at enhancing health worker engagement, efficiency, and patient outcomes, thereby enhancing CHWs' capabilities and community health.

## CCS CONCEPTS

• **Computing methodologies** → **Artificial intelligence**; • **Software and its engineering** → *Software creation and management*; • **Applied computing** → **Health informatics**.

## KEYWORDS

global health, community health, artificial intelligence, reinforcement learning, adaptive interventions

**ACM Reference Format:**
África Periáñez, Kathrin Schmitz, Lazola Makhupula, Moiz Hassan, Moeti Moleko, Ana Fernández del Río, Ivan Nazarov, Aditya Rastogi, and Dexian Tang. 2024. Optimizing HIV Patient Engagement with Reinforcement Learning in Resource-Limited Settings. In *Proceedings of the 7th epiDAMIK ACM SIGKDD International Workshop on Epidemiology meets Data Mining and Knowledge Discovery, August 26, 2024, Barcelona, Spain.* ACM, New York, NY, USA, 6 pages.

## 1 INTRODUCTION

In low- and middle-income countries (LMICs), numerous barriers limit access to quality medical care, particularly in rural areas. These barriers include a shortage of healthcare providers, financial constraints, logistical challenges, and supply chain issues. Also, entrenched behaviors often deter patients from adhering to prescribed treatments and accepting preventive measures such as vaccinations.

The rise of digital health offers new opportunities to address these challenges. Artificial Intelligence (AI) and particularly reinforcement learning (RL) can be transformative in this context, leveraging vast amounts of data generated by users of digital devices to optimize health outcomes. AI can analyze patient behavior, clinical results, care quality, and service demand, predicting future trends and personalizing interventions. Through mobile apps, these adaptive interventions can reach their users and be tailored to individual needs, providing timely support that adjusts to changing circumstances. Furthermore, machine learning (ML) algorithms can predict and manage fluctuations in demand for medical supplies, ensuring efficient resource distribution with timely reminders.

Adaptive interventions and digital health platforms can utilize RL to customize HIV treatment and prevention plans based on real-time patient data, improving treatment adherence and outcomes. Whenever possible, these tools can engage patients directly through mobile devices, offering personalized reminders and educational content that reinforce treatment protocols. Real-time data collection allows for immediate analysis of patient behaviors and treatment effectiveness, enhancing intervention strategies. Additionally, digital health initiatives provide confidential access to healthcare, reducing the stigma associated with HIV and encouraging more

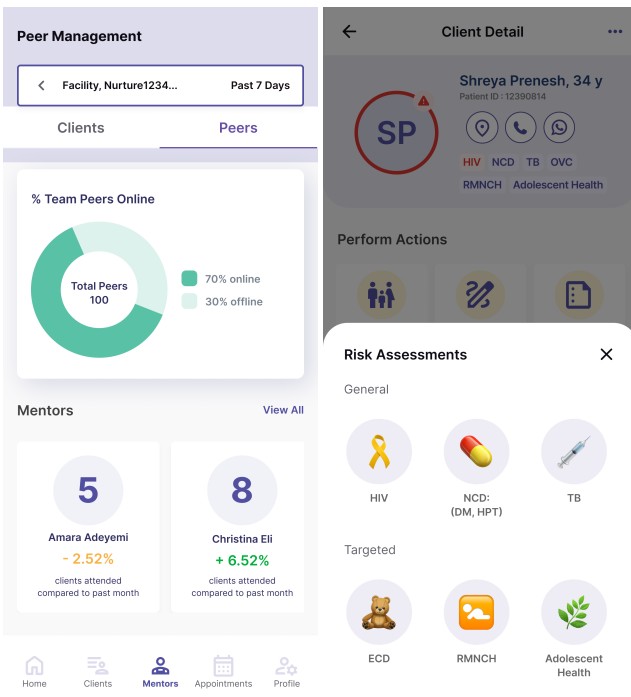

**Figure 1: Screenshots of the AI-optimized mobile health application CHARM (Community Health Access and Resource Management) from mothers2mother organization.**

people to seek treatment. Targeted preventive information and incentives can be directed to at-risk populations through digital platforms, optimizing the reach and impact of HIV prevention efforts.

Healthcare providers often have access to smartphone technology even when the targeted population does not. AI-driven systems provide timely reminders and prediction of the supply chain for antiretroviral or PrEP drugs, ensuring consistent availability to prevent treatment interruptions. Integrated care coordination via digital tools links diagnostics, treatment, and follow-up, managing HIV comprehensively, including its co-infections and co-morbidities. Community health workers (CHWs) in remote areas can gain access to critical support and real-time public health information, enhancing their ability to deliver effective care. These technologies can empower healthcare providers to deliver more coordinated and efficient care, significantly improving patient management and providing patient-specific guidance and clinical decision support.

We believe that adaptive interventions and digital health will be pivotal in transforming HIV care in low- and middle-income countries, making healthcare more responsive and patient-centered. This paper presents the first steps toward this goal of the joint partnership between Causal Foundry (CF),company that builds AI products for healthcare, and mothers2mothers (m2m), a non-profit organization providing community-based integrated primary healthcare services with a particular focus on mother-to-child HIV transmission prevention.

## 1.1 The CHARM App and CHARM Central

CHARM is an advanced Android application designed for peer mentors (m2m's CHWs) and their supervisors to optimize healthcare delivery and streamline administrative tasks. Key functionalities include biometric login for secure access, patient and visit management, risk assessment, service tracking, and a comprehensive referral system that notifies healthcare facilities of new incoming patients to ensure seamless transitions and continuity of care. The platform also supports managing appointments, allowing CHWs to schedule and manage patient visits efficiently, including reminders and follow-ups when patients do not show up, reducing missed appointments and enhancing patient engagement. Additionally, the system stores and manages detailed patient and family health profiles, enabling personalized care. Figure 1 shows screenshots of the CHARM app.

Supervisors have CHW+ profiles that enable them to efficiently distribute their team's workload and track performance right on the app. Key performance indicators (KPIs) for their team and each of its members are readily available without needing to extract data from any database or fill out paper-based forms.

The app facilitates comprehensive health assessments across various areas, including HIV, tuberculosis (TB), non-communicable diseases (NCDs), adolescent health, mental health and substance abuse, reproductive, maternal, newborn, and child health (RMNCH), and early childhood development. It provides guided risk assessments through questionnaires to detect who needs further support and inbuilt workflows for the services provided to those patients.

CHARM Central (C-Central) is its administrative counterpart and a web platform offering a robust operations panel. Program managers can manage CHWs, including their allocations, timesheets, and allowances. C-Central also allows them to track performance and quality of care and build reports per facility, region, or country, ensuring customization and scalability.

A key feature of CHARM is its ability to use AI to assist CHWs and their managers in their daily tasks, including RL-based patient prioritization (for calls, visits, and referrals) and enhanced decision support. By analyzing data and identifying patterns, the app can issue recommendations to assist CHWs in effectively allocating their time and resources and tailoring their response to each patient's need, ensuring that high-risk patients receive immediate and adequate attention.

The assessments allow CHWs to collect critical health data over time, which can be used to generate individual predictions and tailor interventions to meet specific health needs. By leveraging machine learning, CHARM evaluates various data points, such as patient location, family health status, and current and past assessments, to instantly determine patient risk levels and issue recommendations on what to do next, enabling CHWs to make informed and timely decisions. Similar AI capabilities for C-Central can help identify peer mentors needing support and bring that information to the program manager's attention.

CHARM includes a notification center for critical updates, tools for behavioral nudges, and content-based assistance to provide timely patient advice. Its integration with a Large Language Model (LLM) assistant enhances communication and information retrieval,

helping CHWs with tasks like summarizing patient profiles and connecting with specialists.

## 1.2 Community Based Digital Health

Community Health Workers (CHWs) are crucial in HIV prevention, diagnosis, and management. As essential connectors between healthcare facilities and communities, CHWs conduct health screenings and assessments within communities, thereby enhancing access to healthcare for those who may not frequently visit medical facilities. Community-based care minimizes the need for patients to travel for essential services. It also reduces the operational burden on healthcare facilities, allowing them to focus on the patients needing higher levels of care and in-depth medical reviews.

Digital tools like the CHARM application aim to improve CHW engagement and efficiency, increasing the volume and quality of community-based health screenings and services. AI tools further augment CHW efficiency by supporting decision support, capacity building, logistics, and connectivity with healthcare facilities. These tools help CHWs identify patients at risk of discontinuing treatment, needing additional support, or requiring referrals. CHWs benefit from digital platforms that integrate patient information, clinical workflows and guidelines, supply orders, referrals, and communication with their patients, peers, and supervisors, offering guidance and incentives to improve care quality.

Empowering Peer Mentors as CHWs with intelligent digital solutions assists them in reaching the most vulnerable populations and decentralizing HIV care, making it more accessible, and ensuring continuous treatment. This strategy optimizes healthcare resource allocation and maximizes the impact of HIV prevention and treatment adherence among underserved populations.

## 1.3 An AI Platform for Adaptive Intervention

The CF platform integrates into existing digital tools to collect extensive time-varying datasets, synthesizing behavioral, clinical, and contextual information to craft and implement personalized interventions based on individual predictions, such as nudges and in-app rewards. It is represented schematically in Figure 2. It features a structured front-end for user interaction, enabling data analysis, model management, and the creation and testing of interventions. The backend serves as the operational core, organizing data for ML models, ensuring data security, and facilitating real-time interventions. Additionally, API and SDK bridge digital tools with the platform, optimized for various mobile devices in LMICs, ensuring compliance with the General Data Protection Regulation (GDPR) and the Health Insurance Portability and Accountability Act (HIPAA). Section 2 details the methodological and algorithmic choices underlying its adaptive intervention capabilities.

## 1.4 Current Development Status

CF has developed the CHARM app and C-Central in partnership with m2m as the all-in-one tool for peer mentors. Its current version includes patient registration, appointment management, risk assessments across the conditions currently managed by peer mentors, and guidance through the services provided for high-risk patients. Its soft launch for a selected group of peer mentors will take place in August 2024, with its rollout to all peer mentors across the

seven countries where m2m operates happening in phases throughout the following ten months. The development of the tool will continue in parallel to include additional capacity building and communication capabilities. All CF-developed tools are integrated with the CF AI platform (fully operational with multiple partners and in constant development) by design, meaning they have out-of-the-box predictive and intervention capabilities. Experiments will run for different adaptive interventions jointly with the rollout, first focusing on continuous personalized onboarding and support to ensure a seamless transition. Once the adoption is widespread and enough data is collected, the focus will shift to risk profiling and patient prioritization, peer mentor efficiency enhancement, and patient-specific decision support.

## 2 METHODOLOGY

Integration with the CF RL Platform enriches mobile health solutions and healthcare digital tools, particularly the CHARM App and C-Central, with adaptive capabilities. The platform processes incoming logs through specialized functionality and use-case-specific data pipelines, categorizing them into dynamic and static traits that aggregate over time. These traits provide insights into user interactions with in-app content, patient risks, behavior, and clinical evolution. They are used to track behavior, group subjects (CHWs, patients, and facilities), and as statistical and predictive modeling features. The following subsections outline the methodological pieces of our approach in the context of CHARM. A detailed description of the framework in the context of health systems can be found in [27] and more technical details in [13].

## 2.1 Predictive Modeling

Different families of ML predictive and recommendation models are helpful in different contexts for different use cases. For example, survival analysis approaches [14, 19, 25, 34], which model time to an event of interest, can be used to characterize risk and determine who to target with interventions, when and how. Formally, the observed data points $(x_j, t_j, c_j)_{j=1}^m$ are used to estimate the conditional survival function $P(T \geq \cdot \mid X = x)$ of the duration until the event of interest, where $x_j \in \mathcal{X}$ are the input features, $t_j \in (0, M]$ is the censored time-to-event, and $c_j \in \{0, 1\}$ the censoring indicator.

Different subjects and events of interest will be relevant for different use cases. For example, CHW churn or CHW failure to complete a continuous professional development (CPD) module can be used to determine which users need additional support (with app onboarding and capacity building, respectively). Different types of events (becoming an undiagnosed positive, missing an appointment, failing to adhere to treatment) associated with the different conditions (e.g., HIV or TB) can be used to provide a complex patient risk characterization, which can guide the different intervention strategies.

## 2.2 Bandits for Intervention

RL [32] is at the platform's core as it is the ML paradigm best suited to make sequential algorithmic decisions. The platform and methodology outlined above serve as an engine for delivering adaptive interventions and measuring their impact. Both the adaptive delivery

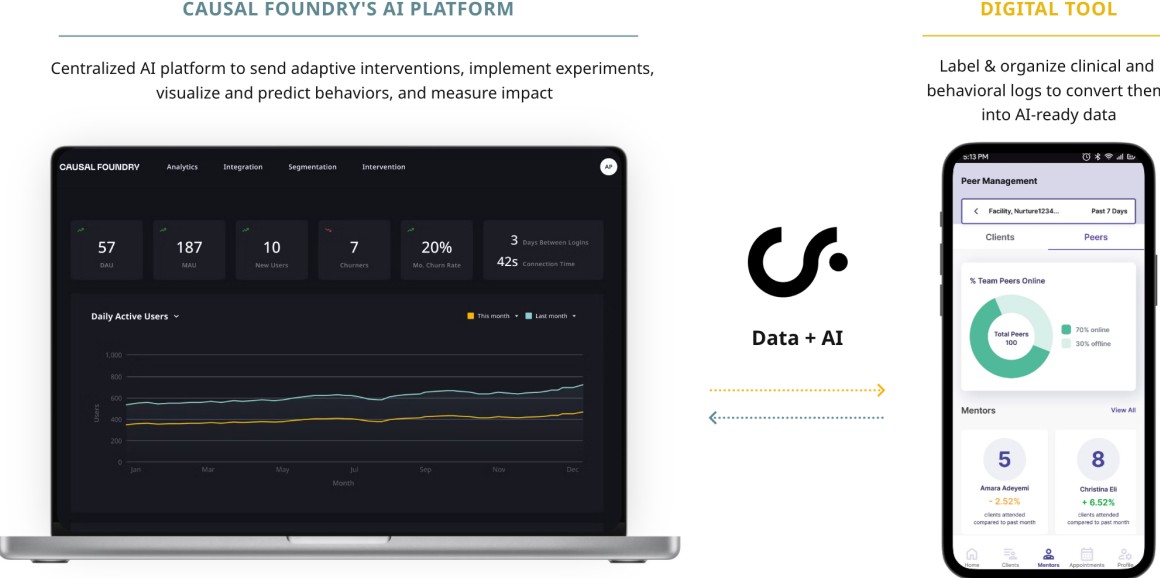

**Figure 2: Reinforcement Learning platform to collect and organize data and deliver personalized, just-in-time adaptive interventions for CHWs and patients, based on contextual and restless bandits, with m2m CHARM application with a focus on diagnosis, prevention, and management of HIV, among other diseases.**

and assignment mechanisms are based on stochastic Multi-armed Bandits (MABs) and Restless Bandits (RMABs).

Bandit algorithms function as online optimization methods that use partial feedback to make sequential decisions, refining their choices based on past interactions to avoid suboptimal outcomes [18]. During each interaction, the algorithm observes a context $x_t$, selects an action $a_t$ based on the previous observed history $\mathcal{H}_{<t} = \{x_s, a_s, r_s\}_{s<t}$, and receives a reward $r_t$, where actions denote intervention decisions, and rewards are based on the traits described above. As the algorithm operates, it aims to minimize the difference between taken actions and optimal actions (regret), assuming the interaction outcome law $r_t \sim p(r \mid a_t, x_t)$ is fixed but unknown, differing from full MDP settings where states evolve under a Markov process [32]. Unlike supervised learning, bandit methods only receive partial feedback, necessitating a balance between taking actions yet to be sufficiently explored and exploiting known successful ones to optimize outcomes. The following subsections detail the various algorithms deployed in production in the RL platform for sequential decision-making.

*2.2.1 Linear Bandits.* A $k$-armed linear bandit uses a linear model for rewards: $r_t = x_t^\top \theta_{a_t} + \varepsilon_t$, where $x_t \in \mathbb{R}^d$ is the feature vector, $(\theta_k)_{k=1}^K \in \mathbb{R}^d$ are the arm reward coefficients, and $\varepsilon$ represents subgaussian noise [7, 21]. Initially, Upper Confidence Bound (UCB) was used for action selection [18, 21] by picking $a_t \arg\max_k x_t^\top \hat{\theta}_k + \text{ucb}_k(x_t; \mathcal{H}_{<t})$, where $\hat{\theta}_k$ is the current $k$-th arm's reward model's coefficient estimate based on $\mathcal{H}_{<t}$. Thompson Sampling allows for

better-defined probabilities for the selected actions using a Bayesian approach [1] with $a_t \sim \mathbb{P}_{\theta \sim \beta_t}(a_t \in \arg\max_k x_t^\top \theta_k)$, where $\beta_t$ is the current posterior belief $p(\theta \mid \mathcal{H}_{\leq t})$. Linear bandits offer sublinear regret guarantees in $T$. However, they are an efficient option for scenarios involving a few key variables influencing optimal actions, such as directing peer mentors to onboarding tutorials based on their experience with the app and simple usage behavior patterns. They are also the best option when the main goal is to learn how these traits affect the best action, such as how the region and age impact the preferred format for tutorials.

*2.2.2 Beyond Linear Bandits.* Linear bandits have limited capacity to represent complex relationships [30]. Other models, such as deep neural bandits, replace the linear reward model with deeper feature extractors [36, 39]. For instance, the stacked neural-linear bandit approach [23] uses a small experience replay queue [11] and a Gaussian-Gamma conjugate prior. Another method models context-action-reward data as a linearized Gaussian state-space model [9], updating the posterior using partial feedback through the extended Kalman Filter [8, 31]. While these may hinder their use for knowledge extraction, they provide more adequate setups for many interventions of interest where the best action depends on complicated ways on many traits. For example, the best service package and schedule (always within approved guidelines) for a concrete patient will depend in complex ways on their characteristics (such as age, location, gender, distance from nearest facility, or family status), known clinical history (such as the evolution of vitals, known

diagnoses, medication prescription and reported adherence, previous pregnancies), previous interactions with peer mentors (such as whether they have missed appointments and reasons given on follow-up, how long they have been in the program, or services they have already received) and predicted risks (of undiagnosed conditions, dropping out of the program, of lack of adherence, complications or psychosocial vulnerability).

*2.2.3 Restless bandits.* A restless bandit setup (RMAB) is closer to the full MDP formulation as it includes a minimal internal state (often binary, such as whether the patient is adhering to treatment) that follows simple (subject-specific) intervention dynamics. RMABs are used for limited resource allocation (i.e., in our framework when the number of available interventions is finite, as are the number of peer mentors and the visits or calls they can make) intending to maximize the number of subjects in a particular state (adhering to treatment in our example). RMABs are modeled as a finite number of concurrent MDPs with incomplete and imperfect information, optimizing a cumulative reward by distributing the available intervention at each decision point. The Markovian dynamics are often also unknown and need to be learned simultaneously. The Whittle index policy is a common heuristic for solving RMABs (a hard combinatorial problem), but it requires indexability to ensure optimal action selection [33]. Equitable RMABs aim to constraint inequities in allocation between different groups by simultaneously maximizing some equity function on the relative differences between group rewards [15]. Examples of RMAB utilization range from wildlife protection [29] to radio spectrum sensing [3] and include many healthcare related applications [2, 4, 5, 16, 20, 22, 24, 26]. They are CF platform mechanisms for sequential resource allocation and can be used for patient prioritization recommendations (for visits and calls) in the context of CHARM.

## 2.3 Experimentation

The RL platform facilitates experiments to measure the impact of interventions using different experimental designs. Assignment to control and treatment groups can be done at individual or cluster levels and can be fully random or adaptive using stochastic MABs [6, 10, 35, 37]. For repeated interventions, intra-subject assignment is an option to increase effective sample sizes in micro-randomized trial designs [17, 28, 38]. See [12, 13] for a discussion of the results of some of the experiments already run with the CF platform.

## 3 SUMMARY AND CONCLUSIONS

CF and m2m have collaboratively developed CHARM, a comprehensive application designed to serve as a job aid and a decision-support tool for peer mentors (m2m's CHWs). This tool is fully integrated with the CF AI platform, endowing it with adaptive capabilities. The CHARM app leverages real-time data, ML predictive models, and bandit-based sequential decision-making to offer personalized, just-in-time interventions to improve patient outcomes and optimize CHWs' efficiency.

The development of CHARM marks a significant step forward in the decentralization and personalization of healthcare services, particularly in resource-limited settings , by leveraging digital health solutions and AI. CHARM enhances peer mentors' capabilities to deliver high-quality care to underserved populations, adapts to patient needs through contextual and restless bandits, and improves healthcare resource allocation for more effective interventions. The collaboration between CF and m2m in creating CHARM highlights the transformative potential of AI-driven digital health solutions, promising significant impacts on global health, particularly in managing and preventing diseases like HIV.

## ACKNOWLEDGMENTS

This work was supported, in whole or in part, by the Bill & Melinda Gates Foundation INV-053824 and by Google.org, AI for the Global Goals Grant Agreement 1007988. Under the grant conditions, a Creative Commons Attribution 4.0 Generic License has been assigned to the Author Accepted Manuscript that might arise from this submission.

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
