# OpenReview forum: "Optimizing HIV Patient Engagement with Reinforcement Learning in Resource-Limited Settings"
_KDD.org/2024/Workshop/epiDAMIK — KDD 2024 Workshop epiDAMIK_

### Official Review · Reviewer_LPyP · 2024-06-23
**Optimizing HIV Engagement with Reinforcement Learning in Resource-Limited Settings- Review**

**Rating:** 2
**Confidence:** 4

**Review:**

Summary:
This paper presents the journey of the development and implementation of the Android application CHARM (Community Health Access Resource Management). This platform supports CHWs in resource-limited settings, particularly in cases of HIV care. The app has features like biometric login, performance tracking, referral management, appointment scheduling, and detailed patient data management, and the authors elaborate on the upcoming reinforcement learning methods. The platform supports comprehensive health assessments for various health conditions, not limited to HIV but also including tuberculosis, non-communicable diseases, and maternal and child health.

Strong points:
1. The implementation of reinforcement learning to prioritize patients ensures that those needing urgent care receive timely attention. This will help identifying patients who critically need care.
2. The paper proposes a general framework that can be applied to a wide variety of diseases, not just HIV.
3. The authors propose that their framework is compliant with the security concerns related to storing personal data and also claim that their platform is scalable.

Weak points:
1. The use of advanced technologies such as reinforcement learning and machine learning may pose challenges in terms of implementation, especially in resource-limited settings. The authors fail to address strategies to mitigate issues that may arise due to this problem.
2. The paper discusses the potential of the platform but may lack detailed evaluation metrics or empirical evidence of its impact on health outcomes.
3. Overall, the authors provide a general overview of their framework but fail to provide particular details about CHARM's development (I understand that recent developments cannot be discussed due to competitive advantage, but the mention of details about the initial operations will be helpful in assessing the practices being applied over the course of time).

---

### Official Review · Reviewer_uudM · 2024-06-30
**Optimizing HIV Patient Engagement with Reinforcement Learning in Resource-Limited Settings**

**Rating:** 0
**Confidence:** 4

**Review:**

Overview: This paper creates a mobile application called CHARM to help community health workers improve HIV care in Africa. The authors overview different bandit methods that can be integrated in CHARM to model events of interest such as predicting which patients/CHWs to nudge, predicting which patients require the most urgent care, etc.

Strengths: This paper has a clear motivation and deployment pipeline. The authors state that the CHARM application has the potential of increasing healthcare access to disadvantaged communities and reducing the operational burden at healthcare facilities. The authors also state that they are working with the mothers2mother organization to deploy the CHARM app.

Weaknesses: The paper currently lays out three different bandit frameworks that can be used to predict events of interest. The main weakness of the paper is that the authors do not provide (1) clear descriptions of how these methods are incorporated into the CHARM app and (2) results that validate the use of these methods. In particular, I advise the authors to focus less on explaining how the methods work, and instead describe what applications the methods are used for, why these methods are suited to predict these events, and empirical validations justifying the use of these methods.

Decision: Due to the lack of a clear methods and results section, I am voting to reject this paper. However, the author’s collaboration with mothers2mother CHWs is very important and impressive work. I wish the authors the best of luck in their partnership!

---

### Official Review · Reviewer_peSW · 2024-06-30
**A promising digital health tool**

**Rating:** 3
**Confidence:** 3

**Review:**

The authors propose a mobile application enhanced by reinforcement learning that aims to improve the engagement of its users to enhance some digital health tools, mostly HIV-related.

Pros:
•	The theme is really important as digital health is very likely to become more important in the foreseeable future.
•	Partnering with an African organization is a step towards creating more equitable and more accurate AI in healthcare products.

Major comments:
•	There is no explanation on how to deal with bias, or dataset shift.
•	It is unclear how this app can be deployed on a massive scale to benefit many people. Many citizens in many countries who could benefit from such applications may not have access to a smartphone for instance.

Minor comments:
•	Please provide a GitHub repository of the codes to allow replication of the analysis, or at least a clear explanation on how to access the models trained, used, and deployed, as well as the application (CHARM) mentioned in the manuscript.

---

### Official Review · Reviewer_nYPW · 2024-07-02

**Rating:** 5
**Confidence:** 4

**Review:**

Summary: The paper proposes a Restless bandit approach for resource allocation for a real-world application of health care management involving allocating Healthcare workers to patients.

Strengths:
1. The paper is well-motivated and the technical aspects are well explained
2. The deployment challenges are insightful

Weaknesses:
1. No technical novelty is methodology
2. Lack of empirical experiments over alternative methods, which maybe hard due to constraints of real-world deployment.